# *Trichosporon asahii PLA2* Gene Enhances Drug Resistance to Azoles by Improving Drug Efflux and Biofilm Formation

**DOI:** 10.3390/ijms24108855

**Published:** 2023-05-16

**Authors:** Xiaoping Ma, Hong Liu, Zhen Liu, Ya Wang, Zhijun Zhong, Guangneng Peng, Yu Gu

**Affiliations:** 1Key Laboratory of Animal Disease and Human Health of Sichuan Province, College of Veterinary Medicine, Sichuan Agricultural University, Chengdu 611130, Chinawangya13570@sicau.edu.cn (Y.W.);; 2College of Life Sciences, Sichuan Agricultural University, Chengdu 611130, China

**Keywords:** *Trichosporon asahii*, *TaPLA2*, overexpression, drug-resistance, cell wall

## Abstract

*Trichosporon asahii* is an opportunistic pathogen that can cause severe or even fatal infections in patients with low immune function. *sPLA2* plays different roles in different fungi and is also related to fungal drug resistance. However, the mechanism underlying its drug resistance to azoles has not yet been reported in *T. asahii*. Therefore, we investigated the drug resistance of *T. asahii PLA2* (*TaPLA2*) by constructing overexpressing mutant strains (TaPLA2^OE^). TaPLA2^OE^ was generated by homologous recombination of the recombinant vector pEGFP-N1-TaPLA2, induced by the CMV promoter, with *Agrobacterium tumefaciens*. The structure of the protein was found to be typical of sPLA2, and it belongs to the phospholipase A2_3 superfamily. TaPLA2^OE^ enhanced antifungal drug resistance by upregulating the expression of effector genes and increasing the number of arthrospores to promote biofilm formation. TaPLA2^OE^ was highly sensitive to sodium dodecyl sulfate and Congo red, indicating impaired cell wall integrity due to downregulation of chitin synthesis or degradation genes, which can indirectly affect fungal resistance. In conclusion, *TaPLA2* overexpression enhanced the resistance to azoles of *T. asahii* by enhancing drug efflux and biofilm formation and upregulating HOG-MAPK pathway genes; therefore, it has promising research prospects.

## 1. Introduction

*Trichosporon asahii*, a widely distributed basidiomycete yeast [1], is a common and important clinical pathogen [2]. The mortality rate of deep fungal disease caused by *T. asahii* is twice that of *Candida albicans* [3], and this fungus is not sensitive to most antifungal drugs. Although initially sensitive to azoles, *T. asahii* has developed resistance due to widespread use [4]. Enhancement of azole effluents is one of the most common mechanisms of fungal drug resistance [5]. *T. asahii* possesses several morphological forms, such as yeast, hyphae, and arthrospores (cell chain) [6]. Biofilm formation by *T. asahii* also enhances its resistance to antifungal drugs [7], and the coexistence of arthrospores and hyphae is conducive to the formation of *T. asahii* biofilm. Owing to these characteristics, *T. asahii* is challenging to treat [8].

Phospholipase A2 (*PLA2*; EC 3.1.1.4), from the esterase superfamily, is closely related to lipid metabolism [9]. Secretory phospholipase A2 (sPLA2) is the most abundant subgroup of PLA2, and it is characterized by highly conserved Ca^2+^-binding sites, His-Asp catalytic dyads, and rich disulfide bonds. sPLA2s are a group of low-molecular-mass enzymes (14–18 kDa) that catalyze the hydrolysis of glycerol phospholipids (hereafter referred to as phospholipids) at the sn-2 position to produce lysophospholipids and free fatty acids [10]. sPLA2s are secreted by cells and require Ca^2+^ to be catalytically competent [11]. Except for the conserved Ca^2+^ binding ring and the active site, the major sequences of the sPLA2 GXII group are significantly different from those of other sPLA2s, and therefore belong to atypical sPLA2s [12]. Despite their low molecular weight, they participate in various biological processes through enzymatic activity and can act as ligands of membrane and soluble receptors [11]. In addition, they participate in regulating physiological functions such as lipid metabolism, signal transduction, and inflammatory responses [13,14,15]. For instance, the sPLA2 identified in *Beauveria bassiana* contributes to lipid droplet homeostasis [16]. *Neurospora crassa* sPLA2 contributes to survival under stress conditions [17]. In our previous study, we screened out the enriched pathway-linoleic acid metabolism based on the transcriptome and metabolome sequencing of *T. asahii* wild-type and resistant strains, and the gene *PLA2* was found to be significantly altered in this pathway. In addition, the metabolite linoleic acid was found to play a part in drug resistance [18]. However, the role of *PLA2* in fungal drug resistance has not been elucidated yet. 

The conventional methods for revealing gene function include deleting a given gene and observing the phenotype of the knockout (KO) mutant. However, this method is limited in some cases, such as when the phenotype cannot be obtained by gene deletion or when the KO method fails [19]. The technology for constructing mutants with genetic defects in *T. asahii* is not yet mature, and therefore, it is challenging to study the effects of specific genes on *T. asahii* [20]. However, artificial gene overexpression (OE) has been applied to various fungi, and it can be used to circumvent the limitations associated with the KO method. Several studies have used OE mutant strains for systematic library analysis and specific gene testing, including those focusing on *C. albicans* [21], *B. bassiana* [16], and *Metarhizium acridum* [22]. In *C. albicans* and other *Candida* species, this method has also successfully identified the biological processes related to fungal pathogenesis, such as biofilm formation, invasive hyphal growth, and drug resistance [23,24,25]. Furthermore, the OE method can complement the exploration and verification of gene functions [26]. In this study, we investigated the construction of *TaPLA2* OE mutants and observed their role in drug resistance.

## 2. Results

### 2.1. Identification of PLA2 in T. asahii

A hidden Markov model (HMM) map of the conserved domain of PLA2G12 (PF06591) was obtained through the Pfam database (http://pfam.xfam.org/ (accessed on 4 August 2022)), and the proteome was screened using an HMM search. Only one gene family member, composed of 144 amino acids, including a typical N-terminal signal peptide (23 amino acids), was found in *T. asahii* (Appendix A). Analysis of the secondary structure of TaPLA2 showed that the isoelectric point of the protein was an unstable hydrophilic protein with a value of 8.59. A CD-search (https://www.ncbi.nlm.nih.gov/Structure/cdd/wrpsb.cgi (accessed on 28 June 2022)) showed that TaPLA2 belonged to the Phospholip_A2_3 superfamily and contained the typical catalytic sites (Appendix A) with a conserved sPLA2 structure [27]. Therefore, *TaPLA2* was identified as belonging to secretory phospholipase A2.

The BLASTP tool was used to search for homologs of this protein in *T. asahii* within the genome database. The compiled phylogenetic tree (Figure 1A) showed that *B. bassiana BBA_07804* and *Cordyceps militaris CCM_07919* were closely related, and they had the highest consistency with the *TaPLA2*.

TMHMM (https://services.healthtech.dtu.dk/service.php?TMHMM-2.0 (accessed on 11 July 2022)) software was used to predict the transmembrane structure of the TaPLA2 protein, and the results showed that it was a non-membrane protein with no corresponding membrane structure. This result was consistent with the prediction that TaPLA2 is a secreted protein (Figure 1B).

### 2.2. Establishing Overexpressed T. asahii Strain

We first amplified a target fragment (approximately 500 bp in size) from the genome of YAN0802 using the primer PLA2-F/PLA2-R (Figure 2A). Fragments were ligated into the vector pEGFP-N1 via homologous recombination. The *TaPLA2* fragment of approximately 500 bp was then amplified by polymerase chain reaction (PCR), and a pEGFP-N1 linear fragment greater than 4000 bp was obtained (Appendix A). The two gene fragments met our expectations, and therefore, the construction of pEGFP-N1-TaPLA2 was considered successful. Vectors expressing the enhanced green fluorescent protein (EGFP) tags were transferred to *T. asahii* via *Agrobacterium* transformation [28]. The EGFP tag was verified by PCR, and a 700-bp band was observed in the TaPLA2^OE^ mutant strain through electrophoresis (Appendix A). Using the same transfer method, the plasmid containing neomycin resistance and the *TaPLA2* coding gene was transferred into *T. asahii*. Therefore, strains growing on the Sabouraud dextrose agar (SDA) medium containing G418 were obtained (Figure 2B). Real-time quantitative reverse transcription-polymerase chain reaction (RT-qPCR) analysis demonstrated that *TaPLA2* expression levels were 118-times higher than those in the wild-type (Figure 2C). The OE strain observed green fluorescence when irradiated with the excitation light (Figure 2D). These results suggested that the TaPLA2^OE^ strain had been successfully constructed and that it would introduce recombinant plasmids into *T. asahii* through gene recombination using *Agrobacterium tumefaciens*.

### 2.3. TaPLA2 Overexpression Enhanced Phospholipase Activity

In this study, the strain’s phospholipase secretion ability was determined by measuring the precipitation circle around the colony on a solid egg yolk medium. When compared with the non-precipitation circle of the wild-type strain, the OE strain showed a clear precipitation circle around the colony, suggesting that OE enhanced the phospholipase secretion of the *T. asahii* strain (Figure 3). The extracellular phospholipase activity of the TaPLA2^OE^ strain was strongly positive according to the method used to calculate the phospholipase activity (Pz value) (Table 1), which proved that the TaPLA2^OE^ strain was successfully constructed.

### 2.4. T. asahii Drug Resistance

Antifungal sensitivity testing was performed on the OE and wild strains using four antifungals: amphotericin B (AMB), fluconazole (FLC), 5-fluorocytosine (5-FC), and voriconazole (VRZ). The broth microdilution data showed increased minimum inhibitory concentration (MIC) values of the overexpressed strain against the azole antifungal agents FLC, VRZ, and 5-FC, particularly against FLC (Table 2). According to the M60 Performance Standards for Antifungal Susceptibility Testing, our results showed that the overexpressed strains were resistant to AMB, VRZ, 5-FC, and FLC.

Characterization of FLC resistance using spot assays confirmed that the overexpressed strain was resistant to FLC. The inhibition concentration of the drug represents the concentration when the size of the colony growing on the SDA medium with the drug is <50% of that of the non-drug colony. The colony formation ability of YAN0802 was significantly inhibited when FLC concentration was 2 μg/mL, whereas the OE of colonies was significantly inhibited on SDA plates at 32 μg/mL FLC (Figure 4).

### 2.5. Overexpression of TaPLA2 Alters Colony Morphology

Colony morphology was observed after 14 days of culture in a standard SDA medium. The overexpressed mutant TaPLA2^OE^ grew slowly, and the observed colonies were opaque with irregular edges and short pili. In contrast, those of the wild-type strain YAN0802 were large, round, and transparent with long mycelia (Appendix A). The colony morphology of the TaPLA2^OE^ strain was similar to that of the *T. asahii* fluconazole-induced resistant strain [29].

### 2.6. Overexpression of TaPLA2 Enhanced Efflux Ability to Promote Drug Resistance

Based on the above results, another drug resistance mechanism, efflux pump expression, was investigated. Genes related to the ATP-binding cassette (ABC) and major facilitator superfamily (MFS) transporters were screened from the transcriptome data for RT-qPCR. As shown in Figure 5, the drug efflux proteins (*PDR11p*, *CDR4*, and ABC transporters) were significantly upregulated, whereas *CDR1* was slightly, but not significantly, upregulated (Figure 5A). In the MFS family, *FUB11* (multidrug resistance protein), *MDR* (multidrug resistance), and *MTP* were all significantly upregulated (Figure 5B). In conclusion, the OE of *TaPLA2* enhanced drug resistance by enhancing strain efflux.

### 2.7. Overexpression of TaPLA2 Enhanced Biofilm Formation to Promote Drug Resistance

In strain YAN0802, hyphae coexisted with arthrospores in abundance at 48 h, while this was observed in the overexpressed mutant TaPLA2^OE^ at 72 h (Figure 6). In addition, the overexpressed mutant strain TaPLA2^OE^ lost its mycelia and reached the yeast phase at a comparatively later stage of culture. The colony morphology of the TaPLA2^OE^ strain was similar to that of the drug-resistant strain, and biofilm formation activity was stronger in the presence of arthrospores and hyphae [6]. These results showed that the ability of *T. asahii* to form biofilms rendered it resistant to antifungal drugs. 

We, therefore, examined the biofilm formation of both the overexpressed mutant TaPLA^OE^ and wild-type strain YAN0802 at 48 h and 72 h, based on the spore, hypha, and arthrospore development. The detection of biofilm formation using 492 nm illumination indicated that the biofilm formation activity of the TaPLA2^OE^ strain was stronger than that of the YAN0802 strain at 72 h (Figure 7).

### 2.8. TaPLA2 Overexpressed Mutant Impaired Cell Wall Integrity

The cell walls of fungi participate in various physiological processes. Cell wall integrity (CWI) is closely related to lipid homeostasis within fungal cells, and lipids are also related to drug resistance [30]. Therefore, maintaining CWI is essential for fungal growth and survival [31]. Sodium dodecyl sulfate (SDS) and Congo red are cell wall inhibitors that can destroy the cell wall and activate stress reactions, including the CWI signaling pathway. The results suggested that the growth of the TaPLA2^OE^ mutant was remarkably inhibited on SDA plates containing SDS and Congo red (Figure 8). This hypersensitivity reaction indicated that OE of *TaPLA2* affected the CWI of *T. asahii*.

To determine the influence of *TaPLA2* OE on *T. asahii* CWI, we examined the expression of genes involved in the synthesis and degradation of chitin. The results showed that the synthesis and degradation genes of chitin were downregulated (Figure 9A). The RT-qPCR analysis also revealed that *ALS* and *GPI*-anchored proteins, associated with cell wall synthesis, were downgraded (Figure 9B). These results showed that the effect of the TaPLA2^OE^ mutant on CWI is related to the downregulation of these genes. The CWI pathway also requires the participation of multiple signaling pathways, such as the HOG-MAPK pathway. We examined the gene expression of this pathway and found that the *HOG* and *MAPK* genes were upregulated (Figure 9C), which indicated that the OE of *TaPLA2* affects CWI but maintains its survival by activating the CWI signaling pathway.

## 3. Discussion

We compared *TaPLA2*-overexpressing strains with their parent strains and analyzed the role of *TaPLA2* in azole drug sensitivity, azole resistance mechanisms, and stress response. sPLA2 comprises a low-molecular-weight Ca^2+^-dependent enzyme with a His-Asp dimeric structure and various isoforms [27]. In our study, we predicted the characterization of the TaPLA2 protein. We conducted a sequence analysis of the TaPLA2 protein and demonstrated that it belongs to the phospholipase A2_3 superfamily with a highly conserved catalytic site but no transmembrane regions. These results were consistent with those reported for other fungi, such as *Tuber borchii* sPLA2 (Appendix A). The *TaPLA2*-overexpressing mutants were constructed through homologous recombination, *A. tumefaciens* transformation, and CMV promoter priming expression. According to the CLSI M60 criteria, the results of MIC experiments showed that the TaPLA2^OE^ mutant has increased resistance to 5-FC, FLC, and VRZ, with the most significant changes in resistance to azole drugs. These results indicate that *TaPLA2* plays a role in the drug resistance of *T. asahii*. The mechanism of increased resistance of the TaPLA2^OE^ mutant to azole drugs is shown in Figure 10.

Our results showed that the TaPLA2^OE^ mutant exhibited the greatest change in resistance to azole drugs, so the known resistance mechanism of azole drugs was studied. Enhanced azole efflux is one of the mechanisms underlying azole resistance in fungi [5]. The expression of efflux-related proteins in the TaPLA2^OE^ strain was detected through RT-qPCR, and the OE of *TaPLA2* was upregulated in the expression of efflux genes in the *T. asahii* strain, which increased azole resistance. 

Biofilms prevent antifungal drugs from entering fungi, which complicates the treatment of fungal diseases [32]. Notably, the ability of *T. asahii* to form biofilms confers resistance to antifungal drugs. *T. asahii* exists in three forms: spores, arthrospores (cell chains), and hyphae [6]. Both arthrospores and hyphae are necessary for biofilm formation, which is enhanced in the presence of arthrospores. In this experiment, the effect of *TaPLA2* OE on biofilm formation activity was measured using an XTT-reduction assay. The results showed that its biofilm formation was stronger than that of the wild-type strain, YAN0802, at 72 h. The growth and development of the overexpressed and wild-type strains were analyzed, and both strains were observed to have hyphae. However, the hyphae of the overexpressed strain were shorter, and it had a greater number of arthrospores compared to the wild-type strain; these factors could provide it with an increased biofilm formation ability.

In other fungi, sPLA2 has been shown to play a role in stress tolerance [33], but its role differs depending on the microorganism. In this study, the OE of *TaPLA2* enhanced the susceptibility of the strain to cell wall stress reagents, such as CR (Figure 8), and maintained cell wall homeostasis at a low level to maintain growth and survival. Chitin is a crucial component of the fungal cell wall and plays a significant role in maintaining cell wall stability. We examined the genes associated with chitin synthesis and degradation; the results showed that both were downregulated. This combination may be the main factor associated with CWI damage. 

Various pathways are involved in the complete construction of the cell wall. In this respect, GPI-anchored proteins are cell surface proteins belonging to different families in different bacteria, and they play pivotal roles in cell wall synthesis [34]. Our results showed that both *GPI* and *ALS* genes were downregulated in the TaPLA2^OE^ strain, and we consider that this may have led to impaired cell wall synthesis in the TaPLA2^OE^ strain. 

The HOG-MAPK pathway is involved in the CWI signaling pathway [35]. In the present study, the expression of *HOG1* and *MAPK* genes was upregulated in the TaPLA2^OE^ strain (Figure 9C); this shows the activation of the CWI signaling pathway, which promotes cell wall repair when stimulated by stress. The maintenance of CWI is a complex process that requires multiple pathways [31]. Our results showed that the OE of *TaPLA2* affected CWI by reducing the expression of chitin synthesis and degradation genes; however, it maintained cell survival by activating the CWI signaling pathway (Figure 9). The downregulation of chitin synthesis genes leads to a decrease in cell wall components, which causes the cell wall to lose its overall structure [36]. Therefore, drugs are more likely to flow out in the case of enhanced efflux, which increases drug resistance; this hypothesis can be verified in subsequent experiments. In addition, the expression of *HOG1* and *MAPK* genes is associated with drug resistance [37], and their upregulation may reduce the sensitivity of *T. asahii* to azole drugs.

In our study, we found that the OE of *TaPLA2* also altered the morphology and growth rate of *T. asahii* (Appendix A). Phospholipase activity is associated with osmotic and oxidative stress, two cellular responses that play key roles in fungal virulence and antifungal sensitivity [38]. The OE of *TaPLA2* enhanced phospholipase activity, and the mechanism of its role as a virulence factor in host pathogenicity can be studied in the future. *PLB1* in *C. neoformans* is located in the cell wall and promotes fungal survival by maintaining the CWI [39]. In this experiment, TaPLA2^OE^ mutant cell wall synthesis slows down, but phospholipase activity is significantly increased, which indicates that the secretion of *TaPLA2* differs from that of other kinds of phospholipase. Subsequent experiments are needed to delineate its secretion process and subcellular localization. In this study, phospholipase activity was enhanced, CWI was impaired, and resistance to azole drugs was enhanced. We believe that this may be due to the slower synthesis rate of chitin in the fungal cell wall, which enhances penetration and facilitates the efflux of azole drugs, thereby enhancing drug resistance.

## 4. Materials and Methods

### 4.1. Strains and Growth Conditions

The *T. asahii* YAN0802 strain (wild-type strain) was isolated from giant pandas and stored at −80 °C in SDA medium supplemented with 20% (*v*/*v*) glycerol.

*E. coli* Trelief^TM^ 5α strains were purchased from TSINGKE (Beijing, China) and cultured with LB medium. Colonies bearing plasmids were selected on LB medium with ampicillin (50 µg/mL). Plasmid extraction from *E. coli* Trelief^TM^ 5α strains was performed by choosing transformants in a 2 mL LB liquid medium with the appropriate antibiotics overnight at 37 °C.

*A. tumefaciens* EHA105, competent for transformation, was purchased from TSINGKE (Beijing, China) and grown on LB plates with rifampicin (50 µg/mL) (Sangon Biotech, Shanghai, China) and kanamycin (50 μg/mL) (Sangon Biotech, Shanghai, China). Transformants were selected and maintained in an induction medium (IM) [40].

The OE plasmid, pEGFP-N1, was synthesized by Honor Gene (Hunan, China). Selection of *T. asahii* containing the recombinant plasmids was performed on SDA medium supplemented with G418 (200 µg/mL) (Sangon Biotech, Shanghai, China).

### 4.2. Construction of Overexpression Strains

After 5–7 days of cultivation, YAN0802 was picked and incubated in SDB broth medium for 48 h at 25 °C with shaking. Subsequently, it was collected and crushed in liquid nitrogen. Total RNA was extracted from YAN0802 suspensions using a *Steady Pure* Universal RNA Extraction Kit (AG, Hunan, China), according to the product description. Reverse transcription was executed using the Evo M-MLV RT Reverse Transcription Kit (AG). TaPLA2 was cloned using PLA2-F and PLA2-R primers (Appendix A) and subjected to bioinformatic analysis. The amplified fragments were ligated and identified using the pclone007 simple vector kit (TSINGKE). PCR products were extracted using a DNA Gel Extraction Kit (TSINGKE).

The construction of the OE strain is shown in Appendix A. TaPLA2 was ligated to the pEGFP-N1 plasmid to form the recombinant plasmid pEGFP-N1-TaPLA2. Enzymes Quick Cut™ *EcoR I* and *Xho I* (Takara, Beijing, China) were used to create common cut sites for TaPLA2 and the vector pEGFP-N1. The *TaPLA2* fragment containing the restriction site was amplified using the primers TY-F/TY-R (Appendix A) and ligated using the ClonExpress^®^ II One Step Cloning Kit (Vazyme, Nanjing, China). After the screening, a single colony was selected and cultured in LB broth containing ampicillin. The recombinant plasmid was extracted for verification using the primers CMV-F/PLA2-R (Appendix A) after shaking the culture overnight.

pEGFP-N1-TaPLA2 was transferred into the YAN0802 strain using the *A. tumefaciens*-mediated transformation method described previously [28]. pEGFP-N1-TaPLA2 was transformed into the *A. tumefaciens* EHA105 strain via heat shock, and transformants were grown on LB containing rifampicin (50 µg/mL) and kanamycin (50 µg/mL). The appearance of transformants was induced in SDA containing G418 (200 μg/mL), and colonies were isolated and cultured in SDA containing G418 (100 μg/mL). The recombinant vector introduced into the strain was validated through PCR using the primers EGFP-F/EGFP-R (Appendix A) and fluorescence observation, and the confirmed strain was named TaPLA2^OE^.

### 4.3. RT-qPCR 

Total RNA was extracted from the fermentation broth of the TaPLA2^OE^ strain using the Steady Pure Universal RNA Extraction Kit according to the product instructions. Reverse transcription was executed using an Evo M-MLV RT Reverse Transcription Kit. Specific primers, YG-F and YG-R (Appendix A), were used to verify whether the OE mutant strain was successfully constructed. The genes and primers for ABC and MFS transporters are listed in Appendix A, and the primers for cell wall-related genes used for RT-qPCR are listed in Appendix A. All primers were designed using Primer 3 Plus (https://www.primer3plus.com/ (accessed on 7 October 2022)) and the Primer Designing Tool (https://www.ncbi.nlm.nih.gov/tools/primer-blast/ (accessed on 7 October 2022)) and synthesized by Sangon Biotech (Shanghai, China). RT-qPCR was performed using a SYBR Green Pro Taq HS premixed qPCR kit (Hunan, China). 18S rRNA was used as the housekeeping gene. The 2^−ΔΔCT^ method was used to calculate the relative multiples of different genes.

### 4.4. Phospholipase Activity

Gokce et al. devised a method to analyze the phospholipase activity within an egg yolk agar medium [41]. The concentrations of the bacterial suspensions were measured using an Ultramicro spectrophotometer (NanoDrop One^C^, Thermo Scientific, Shanghai, China). At 590 nm, the measured value of the control concentration (a value) was 0.075 ± 0.025, and 2 μL of the suspension was added onto the surface of egg yolk agar. Plates were cultured at 37 °C for two weeks [42]. The reference strain, YAN0802, was the negative control. The Pz value was the ratio of the yeast colony diameter to the total diameter of the precipitation zone [43]. The score of Pz values was as follows: 1 = no activity (negative), 0.5–1 = relatively strong activity, and <0.5 = extremely strong activity. This suggests that a lower Pz value indicates higher enzymatic activity.

### 4.5. Characterization of OE Mutant Strains

The wild-type strain YAN0802 and the overexpressed mutant strain TaPLA2^OE^ were punched on the surface of the SDA medium and cultured for 14 days at 25 °C to observe their growth. A copper circle culture suspension was prepared, and spore germination and mycelium development were observed under a microscope.

### 4.6. Drug Sensitivity Test

The wild-type strain YAN0802 and the TaPLA2^OE^ mutant were grown on SDA for 3–5 days at 25 °C. The cells were suspended in SDB liquid medium, and the concentration was adjusted to 1 × 10^6^ cells/mL, which was then used in subsequent experiments.

#### 4.6.1. Antifungal Susceptibility

The antifungal susceptibility was assessed using the broth microdilution method described by Leong et al. [44]. MIC assays were performed in 96-well plates containing RPMI 1640 medium at suitable drug concentrations. Briefly, two-fold dilutions of fluconazole, voriconazole, amphotericin B, and 5-fluorocytosine were prepared, and the final drug concentrations ranged from 256 µg/mL to 1 µg/mL, with 1 × 10^3^ fungal cells in a final SDB volume of 200 µL. Cells were cultured with shaking at 25 °C and monitored after 48 h. The MIC values of the applied drugs were defined as the minimum concentrations that resulted in a growth reduction of at least 50%. Each experiment was performed in triplicate.

The sensitivity of the wild strain *T. asahii* to inhibitory concentrations of fluconazole was compared to that of the overexpressed strain using spot assays [19]. A series of 10-fold-diluted fungal suspensions was prepared using SDB. The colony size was determined by dropping 2 μL of cell suspension at a concentration of 1 × 10^6^–1 × 10^4^ CFU/mL onto SDA plates, and this was cultured for 48 h at 25 °C. Each experiment was performed in triplicate. Stock solution and drug diluent were prepared according to CLSI and EUCAST guidelines [45]. 

#### 4.6.2. Biofilm Assay

Biofilm formation activity was quantified using an XTT reduction assay [38,46], which was conducted as described above [47]. In brief, 100 µL of a cell suspension (1 × 10^6^ cell/mL in SDB liquid medium) was inoculated into each well of a flat-bottomed, 96-well microtiter plate, which was then cultured without shaking at 37 °C for 1 h. The supernatant was removed, and cells were washed with phosphate-buffered saline. A fresh medium was added and incubated without shaking at 37 °C for 24 h. Next, the supernatant was removed and replaced with fresh medium. Then, planktonic cells were removed after incubation for 24 h, and wells were washed thrice with phosphate-buffered saline. XTT and menadione solutions were prepared as described previously and added to each well. Plates were then cultured in the dark for 2 h at 37 °C, and absorbance was measured at a wavelength of 492 nm. The measurements were performed four times for each assay.

#### 4.6.3. Detection of Cell Wall Integrity

To study the influence of *TaPLA2* OE on the cell wall, strains were cultured in SDA medium with several different stress factors, including 0.003% (*v*/*v*) SDS and 100 µg/mL (*m*/*v*) Congo Red. The YAN0802 strain was inoculated into the same medium as the control. A series of 10-fold-diluted fungal suspensions was prepared using SDB. In this respect, 5 μL of 1 × 10^6^ to 1 × 10^4^ CFU/mL cell suspension was dropped on the middle of a plate containing 100 μg/mL Congo red or 0.003% SDS and cultured at 25 °C for two days to observe the colony morphology. The colony diameter was then measured and recorded using the cross method, and each treatment was repeated in triplicate.

### 4.7. Statistical Analysis

Data were analyzed using analysis of variance (ANOVA) in the OriginPro 2019 software (OriginLab Corporation). Significant differences were considered at *p*-values ≤ 0.05 (* *p* ≤ 0.05; ** *p* ≤ 0.01).

## 5. Conclusions

We successfully constructed a *TaPLA2*-overexpressed *T. asahii* strain (TaPLA2^OE^) and found that the TaPLA2^OE^ mutant significantly increased resistance to azole drugs. By studying the resistance mechanism of azole drugs, it was found that the gene expression of efflux pumps such as *MDR* and *CDR* and the formation of biofilms increased, which was the main cause of azole drug resistance. The mutant is sensitive to the cell wall inhibitors CR and SDS, and its growth is slow, which may be due to the slowing of the chitin synthesis rate due to the downregulation of chitin synthesis or degradation genes *GPI* and *ALS*. However, the mutant maintained CWI by activating the HOG-MAPK signaling pathway to ensure normal fungal survival. Given that *PLA2* plays a significant role in many physiological processes within different fungi, an in-depth investigation of *TaPLA2* is necessary, as it has the potential for use in therapeutic medicine. 

## Figures and Tables

**Figure 1 ijms-24-08855-f001:**
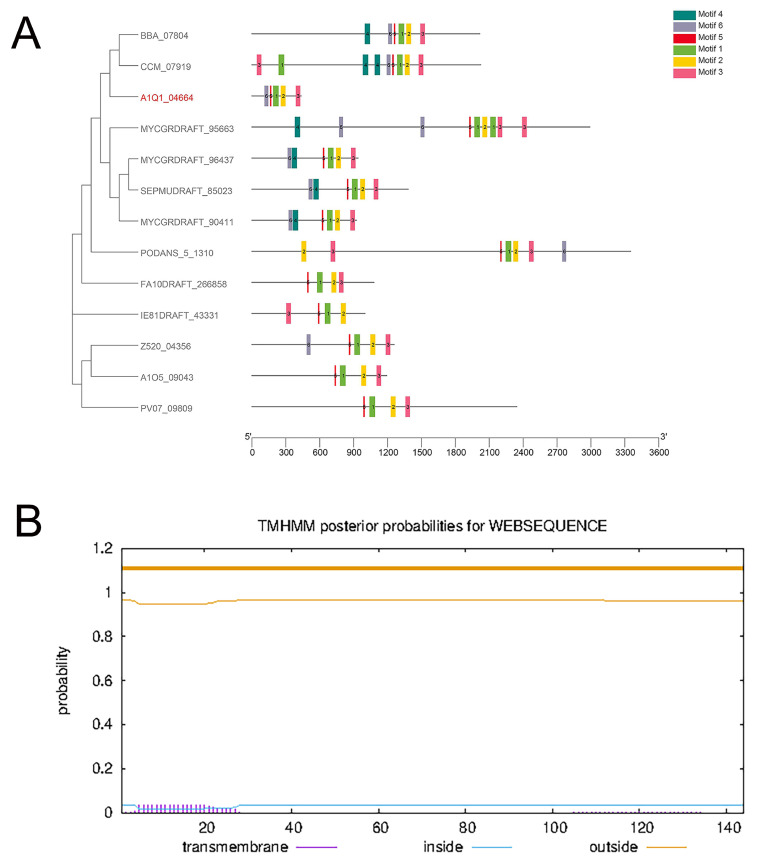
*TaPLA2* structure. (**A**) Phylogenetic analysis of sPLA2s in fungi. (**B**) TMHMM software analysis results showed that TaPLA2 lacks a transmembrane region.

**Figure 2 ijms-24-08855-f002:**
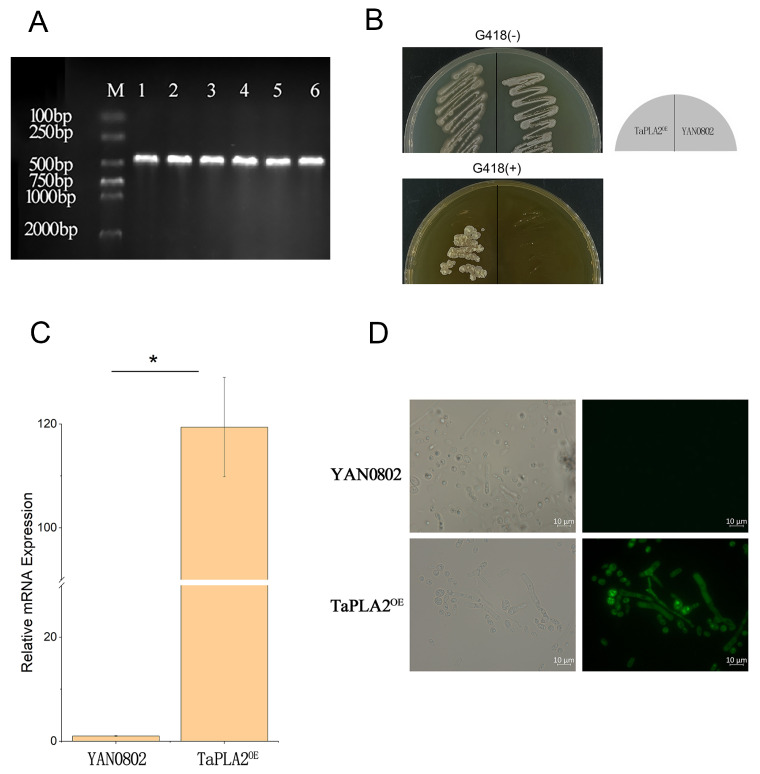
Screening and PCR verification of recombinant strains. (**A**) *TaPLA2* clone fragment (approximately 500 bp). M is the DL2000 DNA marker. Lanes 1–6 show PCR production of the *TaPLA2* gene of YAN0802. (**B**) YAN0802 strain and transformant were strewed on SDA containing G418 (200 µg/mL) and cultured at 27 °C for two days. (**C**) RT-qPCR analysis of *TaPLA2* expression in YAN0802 and transformant. (**D**) YAN0802 and transformant fluorescence were observed by a Zeiss fluorescence microscope. Green fluorescence was observed in the transformants. * *p*-value < 0.05.

**Figure 3 ijms-24-08855-f003:**
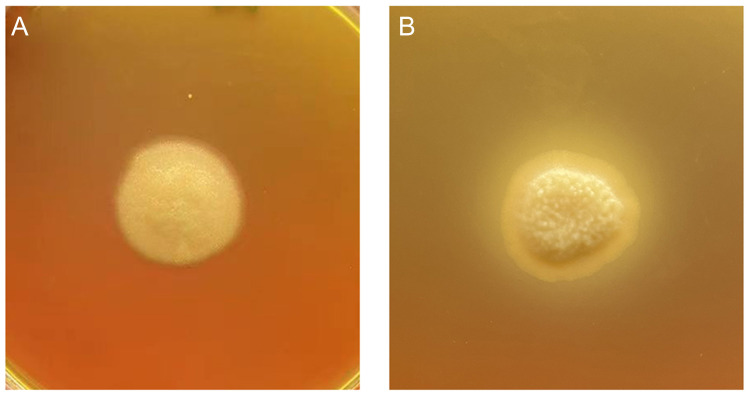
Overexpression of *TaPLA2* enhances extracellular phospholipase secretion. The cells of the strain were spotted on solid egg yolk medium plates and incubated at 37 °C for two weeks. The halo size indicates the relative amounts of secreted phospholipase. (**A**) YAN0802 has no precipitate circle, and (**B**) TaPLA2OE has a clear precipitate circle.

**Figure 4 ijms-24-08855-f004:**
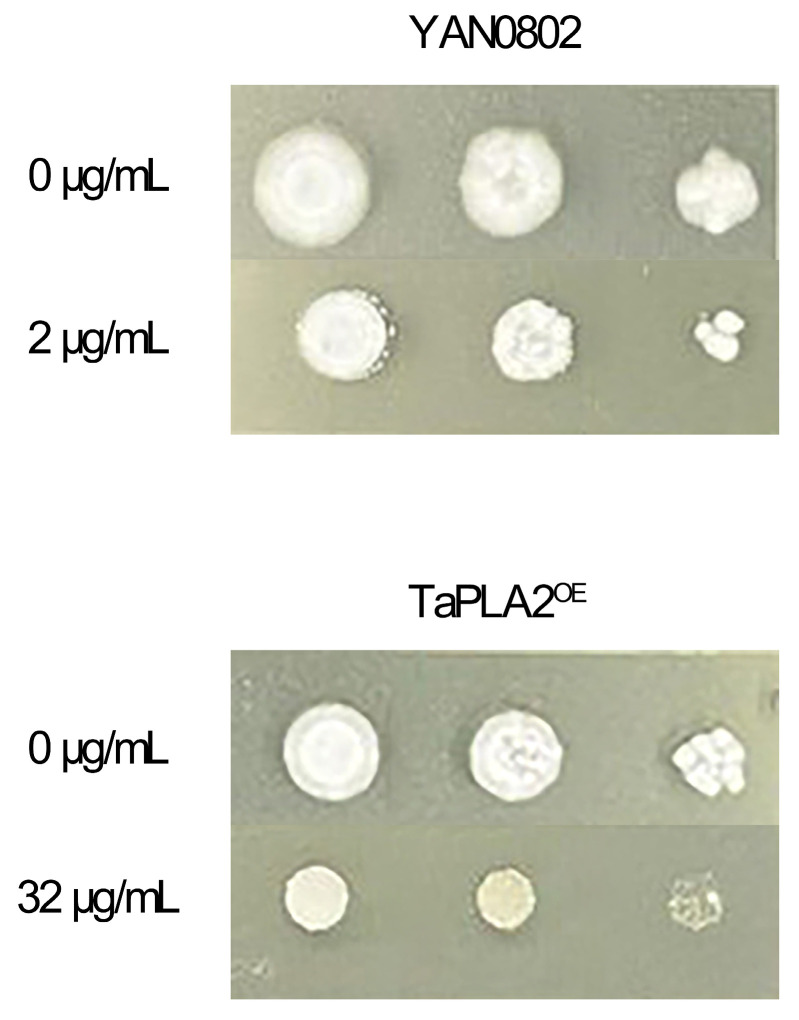
Sensitivity of YAN0802 and TaPLA2^OE^ strains to fluconazole. While considering the 50% of colonies grown on unmedicated SDA as inhibitory concentrations, TaPLA2^OE^ appeared resistant.

**Figure 5 ijms-24-08855-f005:**
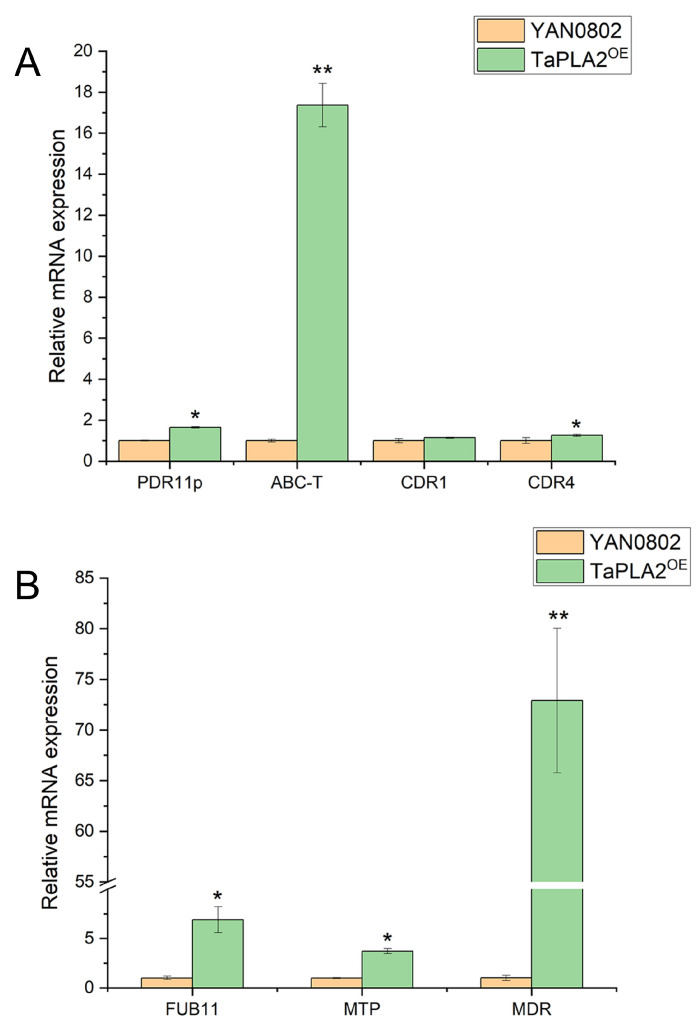
Effect of *TaPLA2* overexpression on efflux pump gene expression. (**A**) Expressions of ATP-binding cassette protein-related genes (*PDR11p*, *ABC-T*, and *CDR4*) were upregulated. (**B**) Expressions of MFS transporter-related efflux genes (*FUB11*, *MTP*, and *MDR*) were upregulated. * *p*-value < 0.05; ** *p*-value < 0.01.

**Figure 6 ijms-24-08855-f006:**
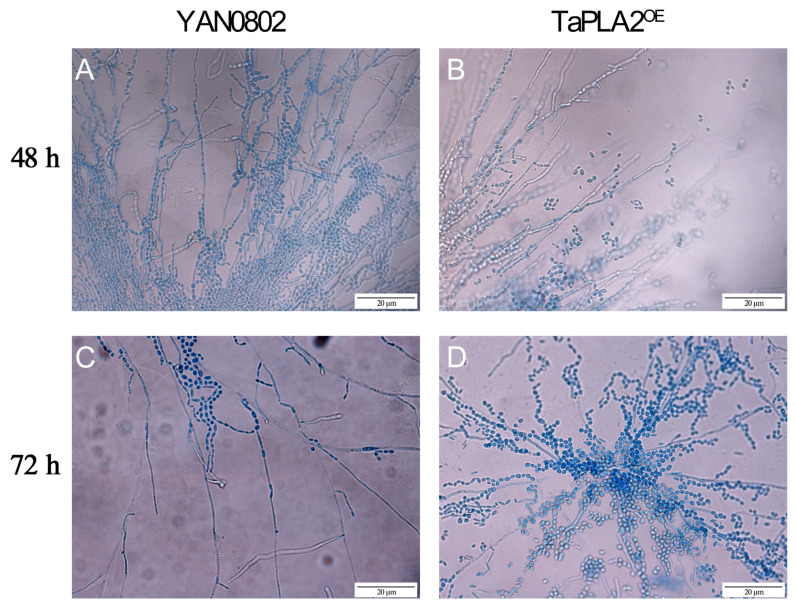
Microscopy images of hyphae, arthrospores, and spores. Arthroconidia and hyphae coexisted in large numbers in YAN0802 and TaPLA2^OE^ strains at 48 h (**A**) and 72 h (**D**), respectively. (**B**) the microscopic state of TaPLA2^OE^ at 48 h, (**C**) the microscopic state of YAN0802 at 72 h.

**Figure 7 ijms-24-08855-f007:**
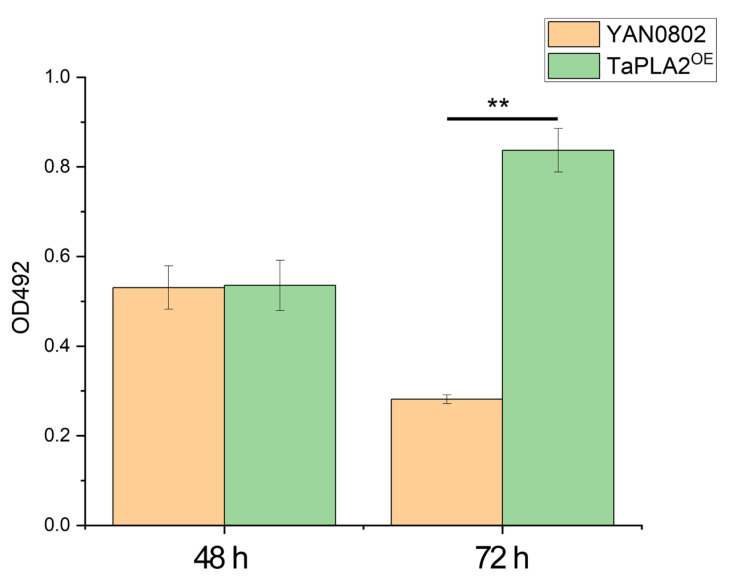
Biofilm formation analyzed by XTT assay. Measurements were conducted four times per condition, and the data are represented as means ± standard deviations. ** *p*-value < 0.01 compared with the YAN0802 strain.

**Figure 8 ijms-24-08855-f008:**
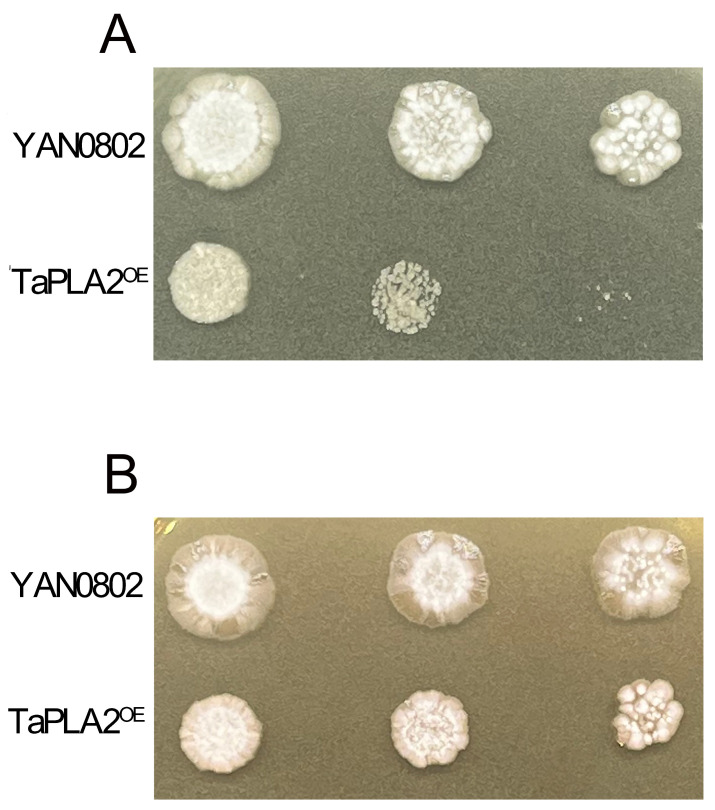
Sensitivity of the TaPLA2^OE^ mutant against cell wall stress inducers. (**A**) Cell suspension spotted on SDA-containing SDS (0.003%) (**B**) and Congo red (100 µg/mL). Each agar plate was cultured at 27 °C for two days.

**Figure 9 ijms-24-08855-f009:**
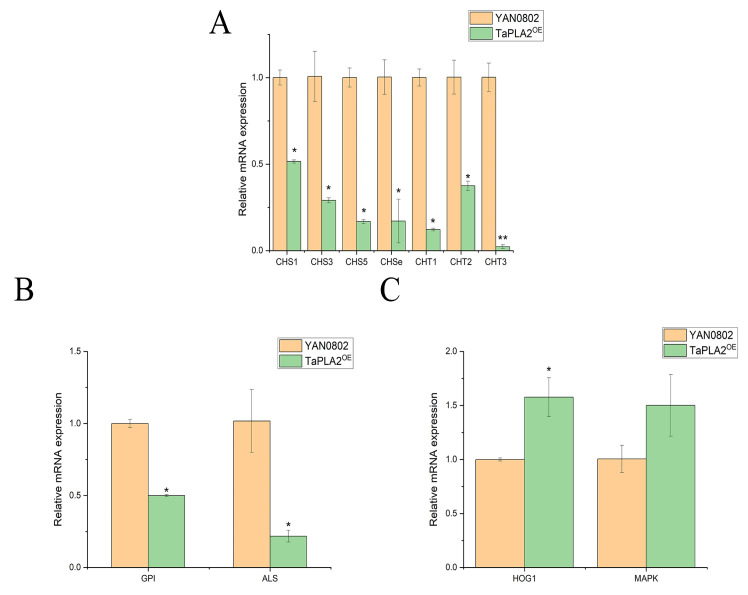
*TaPLA2* affects cell wall synthesis by downregulating the expression of relevant genes at the transcription level. (**A**) Downregulated expression of chitin synthesis and degradation genes. (**B**) Expression of *GPI* and *ALS* in two strains compared using RT-qPCR. (**C**) Expression of HOG1-MAPK genes related to the CWI pathway in two strains compared using RT-qPCR. 18S rRNA transcripts were used as reference genes. * *p*-value < 0.05; ** *p*-value < 0.01.

**Figure 10 ijms-24-08855-f010:**
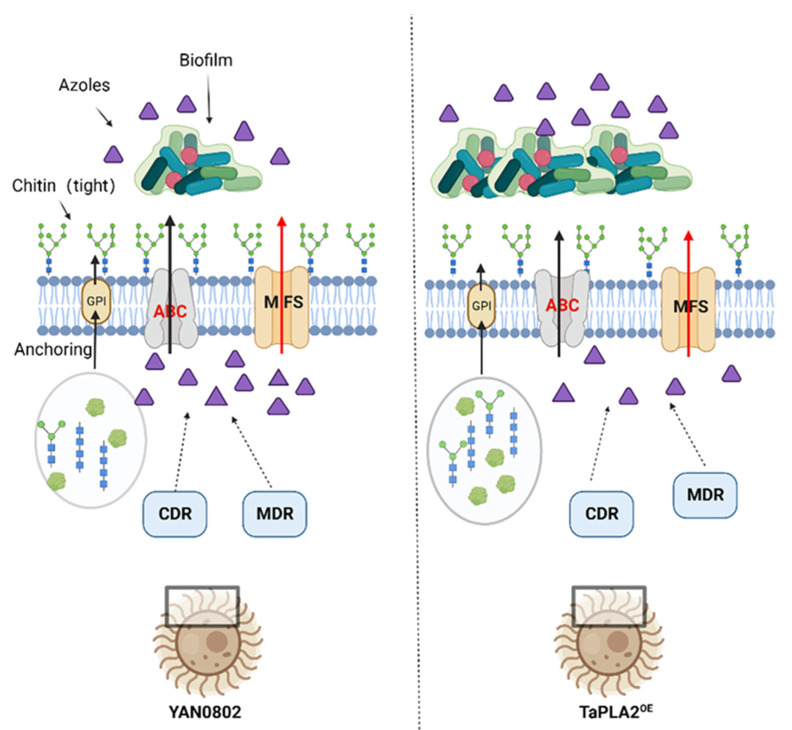
Mechanisms underlying drug resistance in *T. asahii* (created via BioRender.com https://www.biorender.com/ (accessed on 6 February 2023)).

**Table 1 ijms-24-08855-t001:** Comparison of phospholipase activity using Pz values.

	Colony Diameter (mm)	Precipitation Diameter (mm)	Pz (x ± s)
YAN0802	22.83	22.83	1
TaPLA2^OE^	17.35	26.34333	0.658 ± 0.029

**Table 2 ijms-24-08855-t002:** MICs for YAN0802 and TaPLA2^OE^ strains.

Name	MIC (μg/mL)	
	YAN0802	TaPLA2^OE^
5-FC	≥4	≥32
AMB	≥32	≥32
VRZ	0.5	2
FLC	2	≥32

## Data Availability

The data presented in this study are available on request from the corresponding authors. Pfam datasets are available online (http://pfam.xfam.org/ (accessed on 4 August 2022)) and National Center Biotechnology Information. Conserved Domains are available online (https://www.ncbi.nlm.nih.gov/Structure/cdd/wrpsb.cgi (accessed on 28 June 2022)). Primer-BLAST is available online (https://www.ncbi.nlm.nih.gov/tools/primer-blast/ (accessed on 7 October 2022)). Transmembrane structure is available online (https://services.healthtech.dtu.dk/service.php?TMHMM-2.0 (accessed on 11 July 2022)).

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
