# Peer review of "Trichosporon asahii PLA2 Gene Enhances Drug Resistance to Azoles by Improving Drug Efflux and Biofilm Formation"

_ijms, 2023, doi:10.3390/ijms24108855_

Round 1
Reviewer 1 Report
line 19 ... "increase arthropsores": what do you mena? size or the numbers or the realtionhip to mycelia?
line 24: what do you mean with "biofilm actoivity"?
line 24: discrepancy: if phospholipase will destroy cell wall integrity it should promote deactivation of fungal cells and not resistance???
line 39 human cells (irrelvant in this context) or fungal cells?
line 45: why atypical?
line 50: is this relevant in this context dealing with resistance? "lipid droplet homoestasis" within gungal cells or in the viciniy of fungal cells.
(By the way: what is the role of the phospholipase activity in oleaginous yeasts such as Rhodotorula consisting of more than 50% of lipids???)line 53; the induction of inflammatory derivatives of lipds is intersting but this aspect has no influence of these in vitro experiments. this information can be omitted
line 61: defective?????
Table 1: how you explain the small colony size of the mutant? Is evemntually the multiplication process of this mutant impaired??? Hence, the possibly the reduced growth potency may be responsible for the decreased drug susceptibility!!!! The low growth rate is the reason for decerased antimycotoc activity. (also see Fig 8 A+B).
Table 2: Indeed the activity of amphotericin B acting on resting as well as growing fungal cell is not impaired but only the acivity of azoles an dof 5-FC interfereing with the growth of fungal cells. (by the way: is the activity of nitroxoline, which is nomrally active against all Trichosporon spp. tested so far, infuenced by the oveexpression?)
line 161: These data also vote for the assumption that your mutant has acquainted dfect in regeneration!
line 195 and line 198: what is the definition of "biofilm activity"
line 203: do you mean "lipid homeostasis within fungal cells? i.e. the lipoid composition
line 201-209: These results also vozte for the idea that the hyperproducing mutant has experienced during the laboratory manipulation a defect in the replication process. Hence, this strain is powsibl ynot really suitable for antimycotic studies. (I suppose that this mutant has also lost virulence becaus it cannot grow rapidly)
line 216: This finding can slo be interpreted that the mutant is a defective mutant in the growth rate
line 236: this can be seen for example in figure 3 B
line 240; Table 1 shows no higher MIC for ampho!!
line 241: you have forgotten to mention here FLC
line 242: may be the the lower growth rates of the mutant with lower demand of ergosterol is due to the reduced activity of azoles and not the overexpression of phospholipase ???
line 242/45: How overexpression of efflux pumps can explain the the increased rsistance to 5-FC???
line 252:"increases" what do you mean? The biofil as shown i fig. 6 is more compact
line 259: when you have shown these effects such as cell wall stress?
line 269: may be because of the suspected growth defect??
line 271 and 276: you have forgotten to cite here the relevant figures
line 284: where you have depicted the altered morphology?? This would prove that you have a cell-wall deficient mutant??
line 285: again: your data support the impression that your mutant is a cripple with impaired growth properties. Here you should demonstrate a simple growth curve
line 288: this discussion is irrelevant in this context since you have not measured virulence
lines 284-307 can be omitted
line 293: I wonder why in C. neoformans phosphoilipases stabilize the cell wall integrity but obviously not in Trichosporon. (an inhereted defect???)
line 436-445: The findings are possibly correct but the interpretations seems to be inappropriate because a cell wall deficient mutant showing definitive reduced growth rates can easily explain all these presented abnormalities.
Some terms such as "biofilm activity" are not used in a way which could be understood by the reader
Author Response
May 4, 2023
Chidtraporn Lamom
Editor
International Journal of Molecular Sciences
Dear Ms. Chidtraporn Lamom,
Thank you for giving us the opportunity to revise our manuscript (Manuscript ID: ijms-2356426) entitled “Trichosporon asahii PLA2 gene enhances drug resistance to azoles by improving drug efflux and biofilm formation”. The manuscript has been revised according to your suggestions. Our point-to-point responses are attached at the end of this letter.
We hope this revision is satisfactory and we look forward to hearing from you soon.
With kind regards,
Xiaoping Ma and Yu Gu
------RESPONSE TO COMMENTS-----
REVIEWER 1
line 19 ... "increase arthropsores": what do you mean? size or the numbers or the relationship to mycelia?
Response: Thank the reviewers for their constructive comments and suggestions!We have clarified the text to mean the number of arthrospores, line18-19.
line 24: what do you mean with "biofilm activity"?
Response: We have clarified the text to mean “biofilm formation”, line 22.
line 24: discrepancy: if phospholipase will destroy cell wall integrity it should promote deactivation of fungal cells and not resistance???
Response: PLA2 overexpression affects cell wall integrity because of the downregulation of chitin synthesis or degradation genes, but it also activates the HOG-MAPK signaling pathway, which is the signaling pathway used to maintain cell wall integrity/ This may also explain why colony growth slows after PLA2 overexpression, line 265 to 290.
line 39 human cells (irrelvant in this context) or fungal cells?
Response: The text has been clarified to emphasize the previous part. We refer to human cells in the literature, but we study fungi, so we will delete here and emphasize the former part, line 38.
line 45: why atypical?
Response: We meant that the GXII of the sPLA2 family has major sequences that are inconsistent with other SPLA2s except for the typical Ca2+ binding and active sites, so it is an atypical sPLA2.
line 44-46.
line 50: is this relevant in this context dealing with resistance? "lipid droplet homoestasis" within gungal cells or in the viciniy of fungal cells.
Response: sPLA2 has multiple functions, and the text was included to explain several ways in which Spla2 functions and lipids are associated with resistance. The excess text has been removed. line 48-52.
(By the way: what is the role of the phospholipase activity in oleaginous yeasts such as Rhodotorula consisting of more than 50% of lipids???)
- Response: Lipids produced by oleogenic yeast are generally neutral. The correlation between phospholipid metabolism and the anabolism of neutral lipids remains unclear in yeasts. Phospholipase A2 (PLA2) specifically hydrolyzed the sn-2 fatty acyl ester bond of glycerol phospholipid skeleton to produce free fatty acids and lysophospholipids. Although phospholipids comprise a small fraction of total lipids compared with neutral lipids, the hydrolysis of phospholipids by phospholipases played important roles in regulating cellular functions including lipid metabolism by generating various signaling products in fungi. Disruption of the phospholipase PLA2-3 gene (belongs to cPLA2) could effectively improve lipid production in Y. lipolytica. These findings indicated that inactive phospholipase A2 genes might be used to improve lipid production of diverse oleaginous microbes. (PMID: 32648664)
line 53; the induction of inflammatory derivatives of lipids is intersting but this aspect has no influence of these in vitro experiments. this information can be omitted
Response: Some studies have pointed out that there is a certain correlation between lipids and drug resistance. Nonetheless, according to this comment, we have removed the text describing the inflammation, line 51.
line 61: defective?????
Response: What this shows is that there are drawbacks to the knockout method, such as the inability to obtain a phenotype after the deletion of a particular gene. We have changed the word to “limited”, line 58.
Table 1: how you explain the small colony size of the mutant? Is evemntually the multiplication process of this mutant impaired??? Hence, the possibly the reduced growth potency may be responsible for the decreased drug susceptibility!!!! The low growth rate is the reason for decerased antimycotoc activity. (also see Fig 8 A+B).
Response: The concentration of fungal spores dropped on the surface of the SDA medium was the same, and the growth of the overexpressed mutant was slower than that of the wild strain, indicating that the overexpression of PLA2 would affect T. asahii. Therefore, subsequent experiments on the stress of cell wall inhibitor Congo red and membrane disruptor SDS were conducted. The Pz value in Table 1 was obtained by comparing with the diameter of its own colony and the diameter of the precipitating circle. The activity of phospholipase can be reflected according to the Pz value. Fungal growth was inhibited in the medium containing Congo red and SDS, indicating that the fungus was sensitive to cell wall disruptors and that the cell wall or membrane was disturbed. However, the strain maintains the integrity of the cell wall by activating the cell wall integrity signaling pathway, resulting in slow growth.
Table 2: Indeed the activity of amphotericin B acting on resting as well as growing fungal cell is not impaired but only the activity of azoles an dof 5-FC interfereing with the growth of fungal cells. (by the way: is the activity of nitroxoline, which is normally active against all Trichosporon spp. tested so far, influenced by the overexpression?)
Response: The main reason for the amphotericin B resistance values in YAN0802 and overexpressed mutant strains being the same is that Trichosporon asahii is resistant to this drug, so most of the azole drugs are used for the clinical treatment of fungal infections. Nitroxoline is mainly used as an organic synthesis reagent and has no application in antifungal infection.
line 161: These data also vote for the assumption that your mutant has acquainted defect in regeneration!
Response: The overexpression of PLA2 led to slower growth, which may be attributed to the effect of PLA2 on other growth-related genes in fungal cells, line 213-222. This suggests that the gene plays a role in T. asahii. In addition, after we induced resistance to fluconazole in T.asahii, its growth slowed.
line 195 and line 198: what is the definition of "biofilm activity"
Response: A biofilm is an organized community attached to the surface of a fungus and encased in a substrate composed of secreted polymers. Compared with planktic free-floating cells, stroma-coated cells exhibit unique characteristics, including significantly increased resistance to azole and other drugs. For the resistance mechanism of azole drugs, we usually test biofilm formation ability, line 194.
line 203: do you mean "lipid homeostasis within fungal cells? i.e. the lipoid composition
Response: Yes, mitochondrial respiration is important for cell wall integrity, and reactive oxygen species regulate cell wall biosynthesis. In addition, lipid homeostasis can be maintained by calcium ion dependent mitochondrial metabolism, so there is a link between cell wall integrity and lipid homeostasis. Lipid homeostasis is the balance of lipid metabolism, not lipid composition. Line 202
line 201-209: These results also vozte for the idea that the hyperproducing mutant has experienced during the laboratory manipulation a defect in the replication process. Hence, this strain is powsibl ynot really suitable for antimycotic studies. (I suppose that this mutant has also lost virulence becaus it cannot grow rapidly)
- Response: Fungi can lose their growth advantage after drug resistance, and slow growth can even lead to the existence of persistent bacteria. When the fungus is disturbed by external forces, it will adjust its metabolism to adapt to the environment. Fungal resistance is prone to compensatory reactions(PMID: 32824785.The overexpressed mutant may be because the gene change triggers the compensatory mechanism of the strain in other aspects, leading to slower growth of the strain, and the changes in mycelium, spores, and appearance and morphology are similar to those of the induced drug-resistant strains. Therefore, the slow growth of the PLA2-overexpressed mutant may be related to drug resistance, so we believe that it can still be used in the study of fungal resistance. The virulence of fungi generally depends on extracellular secretory enzymes, and phospholipase is one of them.
line 216: This finding can also be interpreted that the mutant is a defective mutant in the growth rate.
Response: The slow growth of fungi is due to downregulated expression of chitin synthesis or degradation genes, which may be a compensatory mechanism caused by the overexpression of PLA2 that makes strains resistant to drugs. Line 213-222.
line 236: this can be seen for example in figure 3 B
Response: Amino acid sequence alignment is shown in Fig. S2B. Fig. 3B in line 134 shows the phospholipase activity of the overexpressed mutant observed on the medium.
line 240; Table 1 shows no higher MIC for ampho!!
Response: I have changed the description in line 241.
line 241: you have forgotten to mention here FLC.
Response: I have added FLC in line 241.
line 242: may be the the lower growth rates of the mutant with lower demand of ergosterol is due to the reduced activity of azoles and not the overexpression of phospholipase ???
Response: FLC and VRZ belong to the azole class of drugs. All antifungal drugs were treated with the same concentration against the wild strain YAN0802 and overexpressed mutant at the same time, and the drug activity was consistent. Only the expression of PLA2 in the overexpressed mutant was inconsistent with that of the wild strain YAN0802, so it could be considered that the drug resistance was caused by the overexpression of phospholipase. Line 241-243.
line 242/45: How overexpression of efflux pumps can explain the the increased rsistance to 5-FC???
Response: Effector pump enhancement is one of the resistance mechanisms of azole drugs. Since TaPLA2OE mutant has the largest change in the resistance value of azole drugs, the resistance mechanism of azole drugs was tested. The antibacterial mechanism of 5-FC is involved in interfering with pyrimidine metabolism, RNA and DNA synthesis and protein synthesis. ( doi: 10.3969/j.issn.1009-0002.2018.06.024) 1.王光裕 ,杨英,车宝泉,李文东,王升启抗真菌药物及其耐药机制研究进展[J],生物技术通讯,2018,29(06):856-860+865.
line 252:"increases" what do you mean? The biofil as shown i fig. 6 is more compact
Response: The biofilm formation was enhanced. Fig. 6 in page 8 shows the microscopic state of spores, mycelia, and articular spores, which have an impact on the formation of biofilms rather than the actual formed biofilms.
line 259: when you have shown these effects such as cell wall stress?
Response: SDS and CR used in cell wall stress tests belong to cell wall/membrane inhibitors and are used to test the integrity of fungal cell walls. These results are shown in line 201-208.
line 269: may be because of the suspected growth defect??
Response: We examined the expression of genes involved in cell wall synthesis or degradation and found that these genes were down-regulated, which affected chitin synthesis or degradation, which may be the reason for the slower growth of our strain. However, cell wall integrity is maintained by multiple pathways. Therefore, further detection of HOG1 and MAPK expression levels of the HOG-MAPK signaling pathway, a cell wall integrity signaling pathway, revealed that their upregulated expression activated the cell wall integrity signaling pathway, thus maintaining the cell wall integrity. Therefore, this does not mean that our mutant was defective in terms of growth, but rather that it grew slowly. Line 213-222.
line 271 and 276: you have forgotten to cite here the relevant figures
Response: We have cited the figures in line280 and 284.
line 284: where you have depicted the altered morphology?? This would prove that you have a cell-wall deficient mutant??
Response: Morphological changes in overexpressed mutants were described in line 162 to 168, and colony morphology is shown in Fig. S4. In our study, we found that the colony growth of the overexpressed mutant slowed and was similar to that of drug-resistant strains induced by this strain. Therefore, it can be assumed that drug-resistant strains will affect growth. In addition, in our study, the downregulated expression of the synthesis or degradation genes of the cell wall component chitin may also cause the slower growth of the strain. However, the upregulated expression of HOG-MAPK related genes in the cell wall integrity signaling pathway maintains the integrity of the cell wall, so the strain is not considered to be defective in terms of the cell wall. Line 213-222.
line 285: again: your data support the impression that your mutant is a cripple with impaired growth properties. Here you should demonstrate a simple growth curve
Response: Our data can only show that the expressed mutant grows slowly compared with the wild strain. Its changes are similar to those of the induced resistance strain, and the overexpression of PLA2 is also a genetic mutation. The strain will undergo certain changes to adapt to this change, which may also be the cause of the slower growth. Most gene changes can affect strain growth. For example, the overexpression of GZF3 in Candida albicans leads to slow growth but increased drug resistance (PMID: 29941885, PMID: 16455487).
line 288: this discussion is irrelevant in this context since you have not measured virulence
Response: We have made adjustments in page 12.
lines 284-307 can be omitted
Response: We have deleted the description related to virulence. Page 12
line 293: I wonder why in C. neoformans phosphoilipases stabilize the cell wall integrity but obviously not in Trichosporon. (an inhereted defect???)
Response: In the example we cited, phospholipase B1 contains anchor protein motifs that enable phospholipase B1 to adhere to the membrane, and studies have shown that phospholipase B1 exists in the cell wall of C. neoformans, so it has the function of maintaining cell wall integrity. The phospholipase A2 we studied was predicted to be secretory phospholipase A2, which is secreted by cells and plays an extracellular role. Its localization is unknown. In our study, cell wall integrity was affected because TaPLA2 overexpression affected chitin synthesis. Line 296.
line 436-445: The findings are possibly correct but the interpretations seems to be inappropriate because a cell wall deficient mutant showing definitive reduced growth rates can easily explain all these presented abnormalities.
Response: In our experiment, it was found that the overexpressed mutant showed slow growth. According to its colony characteristics and mycelium growth, it was found that the growth of the induced drug-resistant strain was similar to that of the drug-resistant strain in the laboratory. In addition, under the condition of slower growth, its sensitivity to antifungal drugs decreased and drug resistance occurred. I think this is because the overexpression of TaPLA2 also falls under the category of genetic mutations, which lead to some adaptive changes in the strain, such as slower growth. In addition, cell wall stress experiments on mutant strains revealed that they were sensitive to cell wall inhibitors. The detection of genes related to the cell wall revealed that the gene for chitin synthesis or degradation was downregulated, which may be the cause of sensitivity to cell wall inhibitors. However, the gene of the HOG-MAPK pathway, a signaling pathway for cell wall integrity, was upregulated. This signaling pathway not only maintains the integrity of the cell wall but also enhances resistance to azole. This series of changes may also cause slower growth. Therefore, we suggest that TaPLA2 overexpression enhances the azole resistance of T. asahii, and this mutant is not a cell wall defective strain. Line 431-442.

Reviewer 2 Report
In the manuscript entitled "Trichosporon asahii PLA2 gene enhances drug resistance by improving drug efflux and biofilm formation and altering cell wall structure", the authors aimed for highly focused studies in a critical field, narrowly specialized and less accessible to readers from other domains.
From the introduction to the conclusions, the same aspects could be observed. The MS addressability is limited to highly specialized professionals in the same field.
The following general comments are available, as follows:
1. The number of references is 53, with only 19 published in the last 5 years ( beginning with 2018).
2. The authors are encouraged to revise all figures in their manuscript and to perform harmonization between the images/graphs/ and external letters.
For example, in Figure 2, the A-D letters' size is around half of a small figure, and the figures with a minimal size do not have an optimal resolution.
The same observations are available for Figure 4. The letters in Figures 5 and 6 are disproportionate to the figure's size. Moreover, Figure 6 could be differentiated into 4 parts, each noted with a suitable letter. Figures 7 and 8 also need suitable revisions.
3. The reviewer encourages the authors to attentively analyze the MDPI instructors for authors, adapting all Figures/Graphs and Tables according to their indications.
4. The most appropriate is Figure 10, placed in the manuscript's final, in the "Conclusion" section. The reviewer opinionates that developing the Discussion section, including Figure 10, is better for illustrating the authors' ideas.
5. The authors are encouraged to check and edit the references in the MDPI style.
The references could also contain the links included in MS (lines 78, 83, 91, 360) for improving its general aspect, highly requested for an MS submitted in a Q1 academic journal with a high impact factor.
Line 431. The reviewer suggests reformulation avoiding repetition.
The reviewer did not observe substantial and frequent language errors.
Author Response
May 4, 2023
Chidtraporn Lamom
Editor
International Journal of Molecular Sciences
Dear Ms. Chidtraporn Lamom,
Thank you for giving us the opportunity to revise our manuscript (Manuscript ID: ijms-2356426) entitled “Trichosporon asahii PLA2 gene enhances drug resistance to azoles by improving drug efflux and biofilm formation”. The manuscript has been revised according to your suggestions. Our point-to-point responses are attached at the end of this letter.
We hope this revision is satisfactory and we look forward to hearing from you soon.
With kind regards,
Xiaoping Ma and Yu Gu
------RESPONSE TO COMMENTS-----
REVIEWER 2
- The number of references is 53, with only 19 published in the last 5 years (beginning with 2018).
Response: Thank the reviewers for their constructive comments and suggestions! I have adjusted the references in page 16 to 19.
- The authors are encouraged to revise all figures in their manuscript and to perform harmonization between the images/graphs/ and external letters.
For example, in Figure 2, the A-D letters' size is around half of a small figure, and the figures with a minimal size do not have an optimal resolution.
The same observations are available for Figure 4. The letters in Figures 5 and 6 are disproportionate to the figure's size. Moreover, Figure 6 could be differentiated into 4 parts, each noted with a suitable letter. Figures 7 and 8 also need suitable revisions.
Response: All the figures you mentioned have been adjusted.
- The reviewer encourages the authors to attentively analyze the MDPI instructors for authors, adapting all Figures/Graphs and Tables according to their indications.
Response: The problem you mentioned has been corrected.
- The most appropriate is Figure 10, placed in the manuscript's final, in the "Conclusion" section. The reviewer opinionates that developing the Discussion section, including Figure 10, is better for illustrating the authors' ideas.
Response: We have revised the Discussion and Figure 10 accordingly in page 11.
- The authors are encouraged to check and edit the references in the MDPI style.
The references could also contain the links included in MS (lines 78, 83, 91, 360) for improving its general aspect, highly requested for an MS submitted in a Q1 academic journal with a high impact factor.
Response: We have modified according to the MDPI reference format.
Line 431. The reviewer suggests reformulation avoiding repetition.
Response: We have revised the statement.
